# The Nicotine Metabolite Ratio and Response to Smoking Cessation Treatment Among People Living with HIV Who Smoke in South Africa

**DOI:** 10.3390/ijerph22071040

**Published:** 2025-06-30

**Authors:** Chukwudi Keke, Limakatso Lebina, Katlego Motlhaoleng, Raymond Niaura, David Abrams, Ebrahim Variava, Nikhil Gupte, Jonathan E. Golub, Neil A. Martinson, Jessica L. Elf

**Affiliations:** 1Department of Environmental and Radiological Health Science, Colorado State University, Fort Collins, CO 80523, USA; chukwudi.keke@colostate.edu; 2Africa Health Research Institute, Somkhele, Myeki 3935, South Africa; limakatso.lebina@ahri.org; 3Perinatal HIV Research Unit (PHRU), University of the Witwatersrand, Johannesburg 2193, South Africa; ptg5@cdc.gov (K.M.); martinson@phru.co.za (N.A.M.); 4School of Global Public Health, New York University, New York, NY 10003, USA; niaura@nyu.edu (R.N.); david.b.abrams@nyu.edu (D.A.); 5Klerksdorp Tshepong Hospital Complex, Matlosana 2574, South Africa; variava@worldonline.co.za; 6Department of Infectious Diseases, Johns Hopkins School of Medicine, Baltimore, MD 21205, USA; ngupte1@jhmi.edu (N.G.); jgolub@jhmi.edu (J.E.G.); 7Johns Hopkins University Center for TB Research, Baltimore, MD 21231, USA

**Keywords:** nicotine metabolite ratio, smoking cessation, HIV, biomarker, nicotine replacement therapy, behavioral counseling

## Abstract

The nicotine metabolite ratio (NMR) has been informative in selecting treatment choices for nicotine dependence and increasing treatment efficacy in Western settings; however, the clinical utility of the NMR among smokers in low-resource settings remains unclear. Prospective analysis was conducted using data from a randomized controlled trial of smoking cessation among adults living with HIV, to examine the association between the NMR and response to smoking cessation treatment. NMR was assessed using bio-banked urine samples collected at baseline. Self-reported smoking at 6 months was verified using a urine cotinine test and exhaled breath carbon monoxide (CO). We found no associations between the NMR and smoking abstinence (adjusted risk ratio (aRR) = 0.82; 95% CI: 0.45, 1.49; *p* = 0.53). No evidence of effect modification by treatment conditions was observed on the multiplicative scale (aRR = 1.17; 95% CI: 0.32, 4.30; *p* = 0.81) or additive scale (adjusted relative excess risk due to interaction (aRERI) = 0.10; 95% CI: −1.16, 1.36; *p* = 0.44). Our results suggest that the NMR may not be a viable approach for selecting smoking cessation treatment in this setting, given the minimal variability in our sample and racial/ethnic makeup of this population.

## 1. Introduction

Sub-Saharan Africa shoulders a disproportionate burden of the global HIV epidemic, with South Africa alone having over 7.6 million people living with HIV (PLWH) [1]. Black South Africans face the greatest impact of HIV, partly due to the lasting effects of apartheid, and are a vulnerable subpopulation [2]. Advances and improved access to antiretroviral therapy (ART) have dramatically improved the longevity of PLWH [1]; however, more people will lose life years from comorbid conditions for which smoking plays a major contributing role [3,4,5]. Evidence suggests that smoking triples the odds of developing active TB [6,7] among PWH compared to their seronegative peers. Additionally, smoking increases the risk of COPD [8], myocardial infarction [9,10,11] and cardiovascular disease among PLWH compared to those without HIV [12].

Research shows that smoking prevalence is two to three times higher among PWH compared to their HIV-seronegative peers in most settings [13]. Among South African adults with HIV, smoking prevalence estimates range from 24% to 74% [6,14,15,16], much higher than 25% in the general population [17]. Despite the high burden of tobacco use in this setting, evidence-based smoking cessation strategies for PLWH are lacking. Intensive behavioral counseling (BC) and combination nicotine replacement therapy (cNRT) have been tested among PWH in South Africa; unfortunately, these interventions have been ineffective in more than 85% of smokers in this setting [18]. Therefore, there is a need for effective strategies to improve smoking cessation in this highly important subpopulation.

The nicotine metabolite ratio (NMR) is a well-established biomarker that characterizes the rate of nicotine metabolism [19,20,21,22]. The NMR is a ratio of the two primary metabolites of nicotine, 3′-hydroxycotinine (3HC) and cotinine (COT), metabolized by the hepatic enzyme CYP2A6 [19,20,21,22]. Variability in the NMR has been found by ethnicity, with smokers of African descent having lower NMR values than their Caucasian counterparts [23]. Further, males have been shown to have lower NMR values compared to females [24]. High NMR levels represent faster nicotine metabolism [19,20,21], and are associated with increased tobacco smoking, greater difficulty in quitting [19,20,21], severe nicotine withdrawal symptoms [25,26], and decreased efficacy of transdermal nicotine patches [26,27,28,29,30,31,32]. The NMR has demonstrated clinical utility in selecting treatment choices for nicotine dependence and increasing treatment efficacy for smokers in the general population [26,27,28,29,30,31,32] and among PLWH [33]. However, these studies have been conducted in Western settings; therefore, the generalizability of these findings to important sub-groups of smokers, such as PLWH in a low-resource setting like South Africa, remains unclear. Understanding whether the NMR will be a viable option for treatment selection among South African adults with HIV has important implications for optimizing cessation interventions for PLWH who smoke in low-resource settings. The primary aim of this study is to examine the association between the NMR and response to smoking cessation treatment among PLWH who smoke in South Africa.

## 2. Materials and Methods

### 2.1. Population

This study used data from a randomized controlled trial for smoking cessation among PLWH in South Africa [18]. In the primary trial, PLWH who smoke were recruited from three primary health clinics in Matlosana, a peri-urban subdistrict located in the Northwest Province in central South Africa, with an estimated population of 398,676 [34] and 30% of antenatal women living with HIV [35]. Eligible participants were HIV-positive as verified by medical record review, ≥18 years of age, self-reported current daily smokers, biochemically verified using urine cotinine tests (≥0.4 micrograms per milliliter (μg/mL) or ≥30 nanograms per milliliter (ng/mL) for participants with tuberculosis (TB) considered as current smoking) and a carbon monoxide (CO) breath test (≥7 parts per million (ppm) considered as current smoking); agreed to participate and anticipated to attend one of the recruitment clinics for ≥6 months; and willing to set a quit date within two weeks after baseline screening. Exclusion criteria included suffering from any health condition that would prevent the use of NRT, current use of smokeless tobacco products, electronic cigarettes, or other smoking cessation treatment, or pregnancy.

The University of Witwatersrand Ethics Committee, the South African Health Product Regulatory Authority, and the Johns Hopkins University Institutional Review Board approved study procedures for the primary randomized controlled trial. The Colorado State University Institutional Review Board approved the current analysis of urine samples. All participants in the randomized control trial provided written informed consent.

Study procedures have been previously described elsewhere [18]. Briefly, eligible participants were randomized to receive either behavioral counseling alone (BC) or BC plus combination nicotine replacement therapy (nicotine patches and gum, cNRT). Regardless of treatment conditions, all participants received advice to quit smoking and self-help materials in a standardized fashion. In-person intensive BC (~45 min) was administered in the clinic at baseline, 2 weeks, 1 month, 2 months, and 3 months, following the South African tobacco smoking cessation clinical practice guideline [36], adapted from the National Cancer Institute’s (NCI) recommended 5As approach [37]. Participants also received additional informational materials from the South African National Council Against Smoking (NCAS). Those randomized to the BC + cNRT condition received a similar BC scheme and an additional 10-week course of nicotine patches and nicotine gum.

### 2.2. Assessment

Self-reported questionnaires administered at baseline were used to collect information on sociodemographic characteristics (e.g., age, gender, socioeconomic status, etc.), basic clinical assessment (e.g., height, weight, etc.), and smoking history (e.g., years of smoking, quit attempts, second-hand smoke exposure, etc.). Smoking behavior variables, such as the number of cigarettes per day and time to first cigarette after waking, were used to estimate the Heaviness of Smoking Index (HSI) as a measure of nicotine dependence [38]. The Wisconsin Nicotine Withdrawal Scale (WNWS) [39] was used to assess nicotine withdrawal symptoms at 6 months, and items were summed to yield 7 primary scales. Participants also self-reported the frequency of alcohol and marijuana use; the CAGE Substance Abuse Screening Tool [40] was used to assess alcohol misuse. Total COT (COT_T_) and total 3HC (3HC_T_) levels were measured in bio-banked baseline urine samples using the liquid chromatography–tandem mass spectrometry (LC/MS/MS) [41] method. The urine NMR was then estimated as a ratio of 3HC_T_/COT_T_. Currently, there is no established cut-point to classify NMR levels; thus, to account for the range of NMRs in our sample, we used the cut-point of the 4th quartile (≥0.3174 ng/mL) to create a binary variable (slow vs. normal metabolizers), consistent with methods in the previous literature [26,30,42]. Smoking status was assessed at 6 months; abstinence was defined as self-reported smoking, biochemically verified using urine cotinine tests and exhaled breath CO. Participants who were lost to follow-up at 6 months were deemed to have continued smoking. Secondary outcomes included changes in self-reported nicotine withdrawal, urine cotinine, and exhaled breath CO at 6 months.

### 2.3. Statistical Analysis

We compared sociodemographic, smoking, and clinical characteristics across smoking abstinence at 6 months using the Chi-squared (ꭓ^2^) test and Wilcoxon rank-sum test, as appropriate. A univariate modified Poisson regression examined the association between the NMR and 6-month abstinence. Potential confounding variables selected *a priori* based on our knowledge of the association and existing literature were adjusted for in the multivariable model. Risk ratios (RRs) and their corresponding 95% confidence intervals (CIs) were reported in both univariate and multivariable models. We examined whether the measure of association was modified by treatment conditions on the multiplicative and additive scale [43]. Effect modification on the multiplicative scale was examined by introducing a product term between the NMR variable (slow vs. normal) and treatment condition in a separate model. The relative excess risk due to interaction (RERI) parameter [44] was used to examine effect modification on the additive scale. We examined the relationship between the NMR and changes in nicotine withdrawal, urine cotinine, and exhaled breath CO at 6 months using linear regression models. All statistical analyses were conducted using R software (version 4.2.1). Tests presented were two-sided, and a *p*-value of ≤0.05 was considered statistically significant

## 3. Results

A total of 561 participants were enrolled in the primary randomized controlled trial. Of this, urine NMR was assessed in 437 participants and included in the final analysis after excluding those with unavailable urine data and those who were deceased during follow-up. There was no significant difference in sociodemographic, tobacco use, or clinical characteristics between our final sample and those excluded from this analysis (Appendix A).

The median (IQR) age of participants was 38 (31, 45) years, and most participants were males (*n* = 342; 78%) (Table 1). Participants reported smoking a median of 10 cigarettes per day (IQR: 5, 15), and the median (IQR) NMR value was 0.312 ng/mL (0.31, 0.32), with most participants (*n* = 328; 75%) classified as slow metabolizers. Among those on ART, only 35 (14%) reported using efavirenz. In the univariate analysis, we found moderate associations between sociodemographic or smoking characteristics and 6-month smoking abstinence (Table 1). Individuals were more likely to be abstinent if they were older (≥45 years; risk ratio (RR) = 2.53, 95% CI: 1.14, 5.61; *p* = 0.02) or female (RR = 1.57; 95% CI: 0.93, 2.65; *p* = 0.09). In contrast, those less likely to be abstinent tended to have a moderate to high addiction to tobacco (RR = 0.59; 95% CI: 0.37, 1.03; *p* = 0.06). No association was seen between other substance use or HIV clinical characteristics and smoking abstinence.

In the multivariable analysis, no significant association was observed between the NMR and smoking abstinence after adjusting for gender as a potential confounder (adjusted risk ratio (aRR) = 0.82; 95% CI: 0.45, 1.49; *p* = 0.53) (Table 2). Additional models that included BMI, efavirenz use, and the number of cigarettes smoked per day as covariates were consistent with the primary analysis (aRR = 0.81; 0.45, 1.44; *p* = 0.48) (Appendix A). Analyses of the joint and individual effects of the NMR and treatment conditions suggest no evidence of effect modification by treatment conditions on the multiplicative (aRR = 1.17; 95% CI: 0.32, 4.30; *p* = 0.81) or additive scale (adjusted RERI (aRERI) = 0.10; 95% CI: −1.16, 1.36; *p* = 0.44) (Table 3). Sensitivity analysis that examined the effects of the NMR and gender suggests no evidence of effect modification by gender on smoking abstinence at 6 months (multiplicative scale: RR (95% CI): 0.70 (0.16, 2.99); *p* = 0.63; additive scale: RERI (95% CI): −0.53 (−2.23, 1.18); *p* = 0.73) (Appendix A).

Separate linear regression models that examined the relationship between the NMR and other secondary outcomes showed no associations between the NMR with either change in nicotine withdrawal (β = 1.01, 95% CI: (−1.91, 3.93; *p* = 0.49), exhaled breath CO (β = −1.38, 95% CI: (−4.52, 1.76, 1.01; *p* = 0.39) or urine cotinine (β = 0.98, 95% CI: (−1.18, 3.15; *p* = 0.37) (Appendix A).

## 4. Discussion

The present study examined the association between the NMR and response to smoking cessation treatment among a sample of PLWH enrolled in a smoking cessation trial. Results showed that the NMR was not associated with smoking abstinence at 6 months, and we found no difference in response to either BC or BC + cNRT. Additionally, there was no association between the NMR and change in nicotine withdrawal or any other secondary outcome. Overall, our study does not support the utility of the NMR as a tool for guiding treatment selection for smoking cessation in this setting. This study provides the first empirical evidence on whether the NMR may be a viable option for treatment selection among PWH in any low- and middle-income country or the African region.

Our results align with previous reports in individuals with psychiatric comorbidities [45] and contribute to the literature regarding the equivocal relationship between the NMR and treatment outcomes [46]. However, this tends to differ from several studies that reported significantly lower quit rates [26,28,29,30,32] and less severe withdrawal symptoms [25,26] among individuals with lower NMR values in the general population. Further, in contrast to earlier reports that indicated increased efficacy of transdermal nicotine among smokers with lower NMR levels [26,27,30], we found no evidence of effect modification by treatment conditions on the association between the NMR and successful smoking cessation in our sample.

There are several explanations for the difference in our observed results. First, our sample comprised entirely treatment-seeking Black South Africans, and we have previously shown minimal variability in the NMR in this population [47], whereas existing evidence comes largely from Western settings with more ethnically/racially diverse populations [26,27,30]. Second, our sample included a higher proportion of individuals with lower NMR values compared to those reported in previous studies. We have previously reported a mean NMR value of 0.32 ng/mL in our sample [47], while the mean NMR in previous studies ranges from 0.35 to 0.44 ng/mL [26,27,30]. The racial/ethnic composition of our sample likely explains the lower NMR values, as individuals of African descent on average metabolize nicotine slower than their Caucasian counterparts, primarily due to a higher prevalence of reduced CYP2A6 gene activity among populations of African descent [24]. Additionally, most participants in our study were not on efavirenz-based ART, which has been shown to induce CYP2A6 activity and nicotine metabolism [48]. Third, the proportion of individuals abstinent at 6 months was lower than what was reported in previous studies. Previous studies that evaluated treatment with NRT in the general population reported abstinence rates ranging from 24% to 42% [26,30], whereas in the present study, the abstinence rate at 6 months was 14%, with similar results among those receiving BC with and without cNRT. Our sample was recruited from a Black South African community with low socioeconomic conditions and limited access to tobacco control campaigns and programs [18], which may likely exacerbate difficulty in quitting [49]. Additionally, our definition of abstinence differed from those used in prior studies. In the present study, self-reported smoking at 6 months was verified using biochemical measures, including urine cotinine tests (≥0.4 μg/mL or ≥30 ng/mL for participants with TB) and exhaled breath CO (>7 ppm). In contrast, prior studies often relied on self-report and CO thresholds ranging from 8 to 10 ppm [26,27,29,30]. Using a similar definition would have resulted in an abstinence rate between 32% and 36% in our sample. Lastly, there was no meaningful difference between treatment conditions for either changes in self-reported withdrawal or any other secondary outcome at 6 months. It is worth noting that participants in our sample smoked fewer cigarettes per day compared to those reported in previous studies [22,26], coupled with a high prevalence of marijuana use. These may have limited our ability to assess withdrawal symptoms using standardized instruments in this setting.

Our study has some notable strengths; the prospective design allowed us to investigate the temporal effects of the association between NMR status and smoking cessation outcomes, which has not been previously studied in this population. Additionally, we adjusted for potential confounding variables selected *a priori* based on our knowledge of the association and existing literature, minimizing confounding bias. The findings of this study are not without limitations. First, we evaluated the NMR as a categorical variable (slow vs. normal metabolizers) using the cut-off of the fourth quartile, consistent with methods in the previous literature [26,30,42]. As previously mentioned, there is no established optimal cut-off point to classify the NMR; therefore, we cannot rule out misclassification of the NMR levels, which may likely bias our observed results towards the null. However, we would expect the magnitude of the bias to be minimal given the limited variability in the NMR in this population. Additionally, the NMR values in our study may not fully reflect the rate of nicotine metabolism in this sample/population, given that we evaluated urine NMR using the ratio of 3HC_T_/COT_T,_ which has been shown to correlate poorly with plasma NMR, CYP2A6 enzymatic activity, and nicotine clearance [50]. Second, the participants included in this study were drawn from a clinical trial for smoking cessation with strict inclusion criteria; therefore, our findings are unlikely to be generalizable to all PLWH who smoke. Replication of the present study in a more representative sample of PLWH will be informative. Third, while the primary trial was sufficiently powered to detect a significant effect in treatment outcomes [18], the present analysis was restricted to only participants with available urine NMR data. Based on the available sample size and observed effect sizes, this analysis was likely powered to detect small between-group differences, although these differences may not be clinically meaningful. Fourth, self-reported smoking at 6 months was verified using biochemical measures (urine cotinine and exhaled breath CO); however, we cannot rule out potential misclassification of our outcome. Urine cotinine and exhaled breath CO have a relatively short half-life of 18 h and 2 h, respectively [51]; therefore, individuals who smoked 24 hours prior to biochemical verification may have been misclassified. Further, marijuana use was highly prevalent in our sample, and smoking marijuana produces CO levels similar to tobacco smoking, which may have limited our ability to measure smoking status correctly [51]. Additionally, urine cotinine is limited in its ability to assess smoking abstinence among patients undergoing treatment with NRT, given that cotinine and nicotine are present at the same levels as smokers [52]. Future smoking cessation programs with NRT should consider alternative biochemical verification of smoking status, such as urinary 4-(methylnitrosamino)-1-(3-pyridyl)-1-butanol (NNAL), which is specific to tobacco exposure and offers a longer half-life of 10 to 45 days [51,52]. Lastly, we assessed withdrawal symptoms using a standardized instrument developed among adult smokers from Western settings, which may be less valid in this setting.

## 5. Conclusions

The NMR was not associated with response to smoking cessation treatment among Black South Africans with HIV who smoke in South Africa. Overall, our results suggest that the NMR may not be a viable tool for selecting smoking cessation treatment in this setting or population. Existing research indicates that ART medications such as efavirenz may upregulate the rate of nicotine metabolism through the CYP2A6 activity [48]. However, with the introduction of dolutegravir as the preferred first-line regimen in this setting [53], future studies should clarify how this medication may influence cessation efforts. Also, additional research is needed to understand how the NMR might impact the efficacy of other interventions in this setting.

## Figures and Tables

**Table 1 ijerph-22-01040-t001:** Sample characteristics and their univariate association with smoking abstinence at 6 months.

	Total (*n* = 437)	Continued Smoking at 6-Month Follow-Up (*n* = 381)	Abstinence at 6-Month Follow-Up (*n* = 56)	*p*-Value	Univariate RR (95% CI)	*p*-Value
NMR						
Slow	328 (75)	284 (75)	44 (79)	0.61	REF	
Normal	109 (25)	97 (25)	12 (21)		0.82 (0.45, 1.49)	0.52
Treatment arm						
BC	214 (49)	190 (50)	24 (43)	0.39	REF	
BC + cNRT	223 (51)	191 (50)	32 (57)		1.28 (0.78, 2.09)	0.33
Sociodemographic						
Gender						
Male	342 (78)	303 (79)	39 (70)	0.15	REF	
Female	95 (22)	78 (21)	17 (30)		1.57 (0.93, 2.65)	0.09
Age, median (IQR)	38 (31, 45)	37 (30, 45)	40 (35, 49)	0.05	1.02 (0.99, 1.05)	0.06
Age						
<35	87 (20)	80 (21)	7 (13)	0.06	REF	
30–35	89 (20)	80 (21)	9 (16)		1.26 (0.49, 3.26)	0.63
36–45	152 (35)	134 (35)	18 (32)		1.47 (0.64, 3.36)	0.36
>45	108 (25)	86 (23)	22 (39)		2.53 (1.14, 5.61)	0.02
Schooling						
<12th grade	369 (84)	322 (84)	47 (84)	1.00	REF	
≥12th grade	68 (16)	59 (16)	9 (16)		1.04 (0.53, 2.02)	0.91
Employment						
Unemployed	322 (74)	277 (73)	45 (80)	0.27	REF	
Employed	115 (26)	104 (27)	11 (20)		0.68 (0.37, 1.28)	0.23
Total monthly family income						
≤ZAR 1000	183 (42)	156 (41)	27 (48)	0.38	REF	
>ZAR 1000	253 (58)	224 (59)	29 (52)		0.78 (0.48, 1.26)	0.31
Tobacco use						
Heaviness of Smoking Index						
Low	105 (24)	86 (23)	19 (34)	0.11	REF	
Moderate/high	331 (76)	294 (77)	37 (66)		0.62 (0.37, 1.03)	0.06
Cigarettes per day, median (IQR)	10 (5, 15)	10 (6, 15)	8 (5, 12)	0.22	0.99 (0.94, 1.01)	0.16
Cigarettes per day						
<10 cigarettes	204 (47)	176 (46)	28 (50)	0.69	REF	
≥10 cigarettes	233 (53)	205 (54)	28 (50)		0.88 (0.54, 1.43)	0.59
Quit attempt in the past year						
No	160 (37)	141 (37)	19 (34)	0.76	REF	
Yes	277 (63)	240 (63)	37 (66)		1.12 (0.67, 1.89)	0.66
Exposed to second-hand smoke at home						
No	295 (68)	257 (68)	38 (68)	1.00	REF	
Yes	140 (32)	122 (32)	18 (32)		0.99 (0.59, 1.69)	0.99
Exhaled breath CO (ppm), median (IQR)	15 (9, 22)	16 (10, 23)	10 (6,19)	0.001	0.96 (0.92, 0.99)	0.01
Smokescreen baseline analysis						
Light	272 (69)	237 (70)	35 (66)	0.57	REF	
Moderate	67 (17)	59 (17)	8 (15)		0.93 (0.46, 1.91)	0.84
Heavy	56 (14)	46 (13)	10 (19)		1.39 (0.73, 2.64)	0.32
Other substance use						
Alcohol consumption						
No alcohol misuse	66 (22)	57 (22)	9 (24)	0.76	REF	
Alcohol misuse	236 (78)	209 (78)	28 (76)		0.87 (0.43, 1.74)	0.69
Current marijuana use						
No	134 (58)	116 (57)	18 (64)	0.37	REF	
Yes	97 (42)	87 (43)	10 (36)		0.77 (0.37, 1.59)	0.48
Clinical characteristics						
Current CD4+ T-cell count						
<200 cells/µL	67 (25)	61 (28)	6 (19)		REF	
200-500 cells/µL	132 (49)	112 (46)	20 (62)	0.08	1.69 (0.71, 4.01)	0.23
>500 cells/µL	72 (26)	66 (28)	6 (29)		0.93 (0.32, 2.74)	0.89
Current viral load						
< 200 copies/mL	19 (26)	17 (27)	2 (20)	0.29	REF	
>200 copies/mL	53 (74)	45 (73)	8 (80)		1.43 (0.33, 6.16)	0.63
Efevarenze use						
No	197 (86)	167 (86)	30 (91)	0.35	REF	0.43
Yes	31 (14)	28 (14)	3 (9)		0.64 (0.21, 1.96)	
Current TB						
No	418 (96)	365 (96)	53 (95)	1.00	REF	
Yes	18 (4)	15 (4)	3 (5)		1.31 (0.45, 3.81)	0.61
Current coug						
No	270 (62)	234 (61)	36 (64)	0.78	REF	
Yes	167 (38)	147 (39)	20 (36)		0.89 (0.54, 1.49)	0.68
Body mass index						
Low (<18.5)	136 (32)	120 (32)	15 (30)	0.17	REF	
Normal (18.5–25)	246 (56)	219 (58)	27 (50)		0.93 (0.52, 1.67)	0.82
High (>25)	41 (12)	41 (10)	10 (20)		1.79 (0.89, 3.61)	0.09

**Table 2 ijerph-22-01040-t002:** Multivariable modified Poisson regression of smoking abstinence at 6 months, including NMR and gender variables.

	RR ^a^ (95% CI)	*p*-Value
NMR (Ref. Slow)	0.82 (0.45, 1.49)	0.53
Gender (Ref. Male)	1.57 (0.93, 2.62)	0.09

Measure of effect modification on the multiplicative scale: aRR (95% CI): 1.17 (0.32, 4.30]; *p* = 0.81. Measure of effect modification on the additive scale: RERI (95% CI): 0.10 (−1.16, 1.36); *p* = 0.44. ^a^ Risk ratio adjusted for gender.

**Table 3 ijerph-22-01040-t003:** Modification of the effect of NMR on smoking abstinence at 6 months by treatment arm.

	Slow Metabolizers	Normal Metabolizers	NMR Within Strata of Treatment Condition
		RR ^a^ (95% CI)	RR ^a^ (95% CI)
Treatment arm BC	REF	0.75 (0.28, 2.02)	0.75 (0.28, 2.02)
BC + cNRT	1.29 (0.71, 2.36)	1.14 (0.48, 2.72)	0.88 (0.39, 2.05)

Measure of effect modification on the multiplicative scale: aRR (95% CI): 1.17 (0.32, 4.30]; *p* = 0.81. Measure of effect modification on the additive scale: RERI (95% CI): 0.10 (−1.16, 1.36); *p* = 0.44. ^a^ Risk ratio adjusted for gender.

## Data Availability

The data will be shared upon reasonable request to the corresponding author.

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
