# Peer review of "The Nicotine Metabolite Ratio and Response to Smoking Cessation Treatment Among People Living with HIV Who Smoke in South Africa"

_ijerph, 2025, doi:10.3390/ijerph22071040_

Round 1
Reviewer 1 Report
Comments and Suggestions for Authors
This is a well written manuscript and addresses a clinically significant problem with great importance to public health. The focus on tobacco users living with HIV in Africa is commendable as this is a highly vulnerable population. I have several major concerns that need to be addressed before I can recommend publication:
1) more details on he urinary NMR method are required; which version of urinary NMR was assessed? This is important as each version has a different relationship with plasma NMR, which is more closely related to CYP2A6 activity and the rate of nicotine clearance than urinary NMR. Please see Nicotine metabolite ratio: Comparison of the three urinary versions to the plasma version and nicotine clearance in three clinical studies - PubMed
2) are any of the participants taking efavirenz? There is growing support for efavirenz being an important inducer of CYP2A6 activity. This point is only briefly mentioned in the conclusion. How might efavirenz use alter the results? See Bien-Gund 2022 (PMID: 34879005), Ashare 2019 (PMID: 30946162), Arens 2022 (PMID: 36083509)
3) Unclear why "high vs normal" is used to describe the NMR classifications, when individuals of African descent are more likely than Whites to have low NMR (as the authors also mention).
4) It is unclear which covariates are included in the models presented in Tables 2 and 3. Also, in Table 2 it is indicated that the RR adjusts for sex, but the RR is also pertaining to the Sex term- this is confusing
5) why was sex not included in the model in Table 3? What about baseline Cig per day, nicotine dependence, or age, which are known to affect cessation outcomes?
6) given known sex and gender differences in smoking behaviours and cessation outcomes, can the authors explore potential sex differences in their models?
Author Response
We thank the reviewers for their thoughtful critique of our manuscript. We have carefully considered each comment and hope to have satisfactorily addressed any concerns the reviewers may have had.
- More details on the urinary NMR method are required; which version of urinary NMR was assessed? This is important as each version has a different relationship with plasma NMR, which is more closely related to CYP2A6 activity and the rate of nicotine clearance than urinary NMR. Please see Nicotine metabolite ratio: Comparison of the three urinary versions to the plasma version and nicotine clearance in three clinical studies – PubMed
Author response: Thank you for pointing this out. The method section has been updated to include the urinary NMR method used in this study, which can be found on line 116 in the revised manuscript. We also made additional edits in lines 254-257 that highlight the limitation of the urinary NMR method.
- Are any of the participants taking efavirenz? There is growing support for efavirenz being an important inducer of CYP2A6 activity. This point is only briefly mentioned in the conclusion. How might efavirenz use alter the results? See Bien-Gund 2022 (PMID: 34879005), Ashare 2019 (PMID: 30946162), Arens 2022 (PMID: 36083509).
Author response: Thank you for pointing this out. We conducted further analysis that included efavirenz use as a covariate in the model, which was consistent with the results of the primary analysis. We have made edits in lines 167-170 in the revised manuscript and supplementary table S2.
- Unclear why "high vs normal" is used to describe the NMR classifications, when individuals of African descent are more likely than Whites to have low NMR (as the authors also mention).
Author response: Thank you for this thoughtful comment. We agree that individuals of African descent are more likely than Whites to have lower NMR values. In this study, the "high vs. normal" NMR classification is based on the distribution of NMR within our study population, which is primarily composed of individuals of African descent. However, we recognize that the terms “high” and “normal” may be misleading, particularly given known population-level differences in NMR. To improve clarity, we have revised the terminology throughout the manuscript to refer to “slow” vs “normal” metabolizers, which more accurately reflects metabolic activity rather than relative values.
- It is unclear which covariates are included in the models presented in Tables 2 and 3. Also, in Table 2, it is indicated that the RR adjusts for sex, but the RR is also pertaining to the Sex term- this is confusing.
Author response: Thank you for pointing this out. We have clarified the covariates included in the models by updating the footnotes in Tables 2 and 3.
- Why was sex not included in the model in Table 3? What about baseline Cig per day, nicotine dependence, or age, which are known to affect cessation outcomes?
Author response: We adjusted for gender in the model presented in Table 3, as noted in the footnote (line 192). Additionally, we conducted a further analysis that included gender, baseline cigarettes per day in the model (see line 166); however, the results of this analysis were consistent with the primary analysis. This result was not shown in the current version of the manuscript. For transparency, we have included the results in a supplementary table.
While we acknowledge that age is causally associated with smoking cessation outcomes, we did not include age in the models because it was not a confounder. Based on our review of the literature and the use of a directed acyclic graph (DAG) to inform causal directions, we hypothesized that age was not associated with NMR, our primary exposure of interest, and therefore does not meet the criteria for confounding.
- Given known sex and gender differences in smoking behaviors and cessation outcomes, can the authors explore potential sex differences in their models
Author response: Thank you for your suggestion. We have included a sensitivity analysis to examine potential gender differences in cessation outcomes, which can be found in lines 173-177 in the revised manuscript and Supplemental Table 3.
Reviewer 2 Report
Comments and Suggestions for Authors
I greatly appreciated having the opportunity to read this article of the Nicotine metabolite ratio and response to smoking cessation treatment among people with HIV who smoke in South Africa. The authors present a prospective analysis using data from a randomized controlled trial (RCT) that evaluated smoking cessation treatment (behavioral counseling with or without combination nicotine replacement therapy) among people living with HIV in South Africa.
This study addresses an important gap in the literature by exploring the clinical utility of nicotine metabolite radio (NMR) in a non-Western, low-resource setting. This focus on a vulnerable, under-studied population is highly valuable. The authors used prospectively collected data from a RCT, which is a robust design for examining treatment outcomes. The use of modified Poisson regression, as well as tests for effect modification (both multiplicative and additive), are appropriate for the research questions. Adjusting for potential confounders selected a priori improves confidence in the reported null association.
While the study is sound overall, there are several aspects that merit further clarification or improvement. Overall, this manuscript makes a significant contribution to the literature on smoking cessation. Its rigorous design and comprehensive discussion of limitations substantially enhance its value. With minor revisions to address the points highlighted thereafter, the manuscript will be a strong candidate for publication.
Although the authors note that Antiretroviral Therapy (ART) medications (like efavirenz) may upregulate nicotine metabolism and that future studies should explore the influence of dolutegravir, this point is touched on only briefly. Therefore, expanding the discussion to include a more in-depth consideration of how ART regimens might interplay with nicotine metabolism—and whether stratified analyses by ART regimen type were possible or planned -would strengthen the manuscript. Also, it might be useful to discuss in more detail how the homogeneity in the NMR values (likely due to genetic factors such as reduced CYP2A6 activity) influences the interpretation of the null findings.
A brief discussion on whether the study was sufficiently statistical powered to detect a modest association between NMR and cessation outcomes would provide important context (sample size considerations)
Ensure that all references to study instruments are accompanied by appropriate detail or references for readers who may be less familiar with these measures.
I recommend acceptance of the manuscript pending minor revisions that address the points raised above—particularly the need to clarify the rationale behind NMR categorization, expand the discussion on ART effects and measurement limitations, and comment on the study’s statistical power.
Author Response
We thank the reviewers for their thoughtful critique of our manuscript. We have carefully considered each comment and hope to have satisfactorily addressed any concerns the reviewers may have had.
- Although the authors note that Antiretroviral Therapy (ART) medications (like efavirenz) may upregulate nicotine metabolism and that future studies should explore the influence of dolutegravir, this point is touched on only briefly. Therefore, expanding the discussion to include a more in-depth consideration of how ART regimens might interplay with nicotine metabolism—and whether stratified analyses by ART regimen type were possible or planned -would strengthen the manuscript. Also, it might be useful to discuss in more detail how the homogeneity in the NMR values (likely due to genetic factors such as reduced CYP2A6 activity) influences the interpretation of the null findings.
Author response: Thank you for your suggestion. We were unable to conduct a stratified analysis due to the low proportion of participants using efavirenz. However, we conducted additional analyses including efavirenz use as a covariate in the model, and the results remained consistent with our primary findings. We also revised the discussion section (lines 223–224) to further clarify the role of ART medications in our sample.
Additionally, we believe that the potential influence of genetic factors, such as reduced CYP2A6 activity, on the interpretation of our null findings is appropriately addressed in the third paragraph of the discussion (lines 219–223).
- A brief discussion on whether the study was sufficiently statistically powered to detect a modest association between NMR and cessation outcomes would provide important context (sample size considerations).
Author response: Thank you for your suggestions. We have made some additional edits in the discussion section, which hopefully help clarify this point (lines 262 – 266).
Round 2
Reviewer 1 Report
Comments and Suggestions for Authors
Excellent revision, congratulations.